# Beer-Derived (Poly)phenol Metabolism in Individuals With and Without Metabolic Syndrome: A Comparative Dietary Intervention

**DOI:** 10.3390/molecules30142932

**Published:** 2025-07-11

**Authors:** Daniel Hinojosa-Nogueira, Cristina María Díaz-Perdigones, María José García-López, Ascensión Marcos, María P. Portillo, Rosa María Lamuela-Raventós, Alba Subiri-Verdugo, Esther Nova, Iñaki Milton-Laskibar, Polina Galkina, Francisco J. Tinahones, Isabel Moreno-Indias

**Affiliations:** 1Instituto de Investigación Biomédica de Málaga y Plataforma en Nanomedicina—IBIMA Plataforma, BIONAND, 29590 Málaga, Spain; cristinnemarie@gmail.com (C.M.D.-P.); mari.jose.baza@gmail.com (M.J.G.-L.); alb.sub@gmail.com (A.S.-V.); isabel.moreno@ibima.eu (I.M.-I.); 2Unidad de Gestión Clínica de Endocrinología y Nutrición, Hospital Universitario Virgen de la Victoria, 29010 Málaga, Spain; 3Department of Medicine and Dermatology, Faculty of Medicine, University of Málaga, 29010 Málaga, Spain; 4CIBER Fisiopatología de la Obesidad y Nutrición (CIBERObn), Instituto Salud Carlos III, 28029 Madrid, Spain; mariapuy.portillo@ehu.eus (M.P.P.); lamuela@ub.edu (R.M.L.-R.); inaki.milton@ehu.eus (I.M.-L.); polinagalkina@ub.edu (P.G.); 5Institute of Food Science, Technology and Nutrition (ICTAN), Spanish National Research Council (CSIC), 28040 Madrid, Spain; amarcos@ictan.csic.es (A.M.); enova@ictan.csic.es (E.N.); 6Nutrition and Obesity Group, Department of Nutrition and Food Science, Faculty of Pharmacy and Lucio Lascaray Research Centre, University of the Basque Country (UPV/EHU), 01006 Vitoria-Gasteiz, Spain; 7BIOARABA Health Research Institute, 01006 Vitoria-Gasteiz, Spain; 8Polyphenol Research Group, Departament de Nutrició, Ciències de l’Alimentació i Gastronomía, Facultat de Farmàcia, Universitat de Barcelona (UB), Av. de Joan XXII, 27-31, 08028 Barcelona, Spain; 9Institut de Nutrició i Seguretat Alimentària (INSA), Universitat de Barcelona (UB), 08921 Santa Coloma de Gramanet, Spain

**Keywords:** beer, (poly)phenols, dietary intervention, metabolism, metabolic syndrome

## Abstract

The consumption of low-alcohol fermented beverages has been related to cardiovascular health improvements. Although the underlying mechanism is not completely understood, (poly)phenols have been proposed as one of the mediators. The objective of this study was to evaluate the impact of a controlled intervention with beer on (poly)phenols metabolism in individuals with and without metabolic syndrome (MetS). 20 participants (MetS and control) who consumed a standardized amount of beer during 6 weeks were recruited. Phenolic compounds were assessed in urine. Different changes in phenolic compounds associated with chronic beer consumption were found, particularly related to hesperetin conjugates and to the degradation of phenolic compounds derived from flavonoids and lignans. Noteworthily, MetS and control participants differed in baseline urine phenolic compound profiles and in their metabolization. Significant differences were found in the production and excretion of key (poly)phenols-derived metabolites, such as increased naringenin phase II conjugates in healthy subjects, or increased bacterial flavonoid catabolites. Certain relationships were observed between the phenolic compounds with metabolic and anthropometric variables. These findings suggest that beer-derived (poly)phenols are differentially metabolized according to metabolic-health status, and that they may contribute to certain metabolic health benefits through the modulation of specific metabolic pathways.

## 1. Introduction

Metabolic syndrome is defined as a cluster of cardiometabolic risk factors that includes elevated blood pressure, hyperglycemia, increased waist circumference or abdominal obesity, elevated plasma triglycerides, and low levels of high-density lipoprotein (HDL) cholesterol. The condition is diagnosed when at least three of the following criteria are met: waist circumference > 102 cm in men or >88 cm in women; triglyceride levels ≥ 150 mg/dL (1.7 mmol/L) or specific treatment for this lipid abnormality; HDL cholesterol < 40 mg/dL (1.0 mmol/L) in men or <50 mg/dL (1.3 mmol/L) in women, or a specific treatment for reduced HDL; blood pressure ≥ 130/85 mmHg or the use of antihypertensive medication; and fasting plasma glucose ≥ 100 mg/dL (5.6 mmol/L) or treatment for elevated blood glucose [1,2]. Furthermore, the presence of these factors increases by 5-fold the probability of developing type 2 diabetes, by 2-fold the risk of cardiovascular disease and by 1.5-fold the risk of mortality [2]. The global prevalence of metabolic syndrome is estimated to range between 20% and 40%, with significant variations across different countries. Moreover, an unhealthy lifestyle, including factors such as smoking, low or no physical activity, a poor-quality diet and other elements, is under the main factors that may potentially increase the metabolic syndrome prevalence [1].

Although the pathophysiology of metabolic syndrome is multifactorial, oxidative stress, resulting from an imbalance between oxidants and antioxidants, has been demonstrated to be a pivotal factor in the pathogenesis of metabolic disorders [1,2].

It is important to note the role of dietary compounds in the maintenance of oxidative stress and its associated effects. The evidence indicates that dietary patterns, particularly those comprising elevated contents of fat or added simple carbohydrates, are linked to oxidative damage through the elevation of protein carbonylation and lipid peroxidation products while concurrently reducing the antioxidant defense status [3]. However, these effects can be mitigated by the intake of specific nutrients, including some minerals and vitamins, as well as non-nutrient bioactive phytochemicals such as (poly)phenols [3,4]. Dietary (poly)phenols are molecules with high antioxidant capacity, also being capable of modulating gene expression and modifying epigenetic alterations, which is essential for the prevention of the pathological conditions associated with metabolic dysfunction [4]. Polyphenol-rich foods include a variety of fermented foods, which are notable for their nutraceutical potency [5]. These foods include certain beverages such as wine, beer, or new beverages [5,6,7]. Although these beverages are classified as alcoholic, and thus may have adverse effects when consumed in excess or over an extended period, numerous studies have demonstrated an association between their moderate consumption and an improvement in metabolic syndrome parameters, which is attributed to the presence of (poly)phenols [6]. In this sense, beer is one of the most consumed alcoholic beverages and has a polyphenolic composition comprising compounds such as phenolic acids and tannins, which have high bioactive potential [8].

Therefore, the aim of this study was to evaluate the effect of a controlled dietary intervention, including beer, on (poly)phenols metabolism in individuals with and without metabolic syndrome.

## 2. Results

### 2.1. Characteristics of the Subjects

This study was a randomized open-label crossover trial (ClinicalTrials.gov Identifier NCT05300165) conducted at Virgen de la Victoria University Hospital (Málaga, Spain). Although the main outcomes were published elsewhere [9], data relative to the aim of the current study, to decipher how chronic beer consumption (six weeks, one beer per day) alters the urine (poly)phenols profile in volunteers with and without metabolic syndrome, is currently presented. The composition of the study population was 65% females and 35% males. The mean age of the participants was 45.6 ± 8 years. The analysis of anthropometric parameters revealed statistically significant differences (*p* < 0.05) between the control (C) vs. metabolic syndrome (MetS) groups. It is important to highlight that the study population was constituted by an equal number of individuals in each group. The mean body weight was higher in MetS (93 ± 15 kg) than in C (66 ± 7.6 kg). Similarly, the mean body mass index (BMI) was higher in MetS (34 ± 4.8 kg/m^2^) compared to C (23 ± 2.8 kg/m^2^). The results showed statistically significant differences (*p* < 0.05) between the C and MetS groups for most biochemical parameters irrespective of the time, including blood pressure, fasting glucose levels, and the lipid profile, which were higher in MetS than in C (Table 1). In these parameters, no statistically significant differences were found between the beginning and the end of the intervention. It is noteworthy that no statistically significant differences in nutritional parameters were neither identified during the study, in which a standardized Mediterranean diet was followed. Furthermore, the principal values obtained for dietary intake with regard to energy and macronutrient consumption are presented in Table 1.

### 2.2. Phenolic Profiles of the Whole Population During the Intervention

Urine samples were subjected to high-performance liquid chromatography coupled to high-resolution mass spectrometry for (poly)phenols screening, particularly those related with microbial metabolism. In order to know the variation followed by the phenolic compounds, principal component analysis (PCA) and other variable reduction analyses were performed (Figure 1). Multivariate analysis techniques were employed to reveal a partial yet consistent clustering of samples according to intervention time (baseline vs. end time) and treatment group (C vs. MetS). While the overlap between the time points suggests a moderate change in phenolic compound profiles over a six-week period of daily consumption of a bottle of beer, Procrustes analysis supported this observation with a high disparity value (0.817), indicating a substantial, though not complete, reorganization of the metabolic space. In contrast, a clearer separation between the intervention groups was observed, particularly in the PCA and partial least squares (PLS) projections. A similar trend is observed with t-SNE (T-distributed stochastic neighbor embedding) and isomap analyses, although it is much less pronounced. This finding was further confirmed by a higher disparity (0.923).

In order to know the particular metabolites that significantly changed in the whole population after the beer intervention, different approximations were performed. Firstly, a fold change analysis was assessed. In this manner, we found that a total of 20 phenolic metabolites, 8 down and 12 up, significantly changed (Figure 2). In particular, hesperetin glucuronide, hesperetin sulfate_2, hesperetin sulfate_1, naringenin glucuronide, naringenin, 3-hydroxyphenylacetic glucuronide_3, dihydroresveratrol sulfate_1, sinapic acid glucuronide_1, coumaric acid sulfate_2, hippuric acid glucuronide_1, hydroxybenzoic acid glucuronide_1, and sinapic acid sulfate_1 suffered an increment in their concentrations, while 3,4-dihydroxyphenylacetic acid sulfate_2, chlorogenic acid, hydroxytyrosol sulfate_2, vanillic acid sulfate_2, 3,4-dihydroxyphenylacetic acid glucuronide_1, 3.4-dihydroxyphenylacetic acid sulfate_3, tyrosol sulfate_1, and sinapic acid suffered a reduction.

In order to gain a deeper insight into these data, we performed a paired *t*-test analysis which revealed that six phenolic metabolites changed between times: hesperetin sulfate_1 (*p*-value = 0.002, FDR-*p*-value = 0.121), hippuric acid glucuronide_1 (*p*-value = 0.002, FDR-*p*-value = 0.121), hesperetin glucuronide (*p*-value = 0.003, FDR-*p*-value = 0.121), 3,4-dihydroxyphenylacetic acid sulfate_2 (*p*-value = 0.004, FDR-*p*-value = 0.121), protocatechuic acid glucuronide (*p*-value = 0.008, FDR-*p*-value = 0.199), and hesperetin sulfate_2 (*p*-value = 0.009, FDR-*p*-value = 0.199), where all the metabolites but 3,4-dihydroxyphenylacetic acid sulfate_2 increased their amounts.

Finally, a volcano plot analysis that mixed both statistical analyses used before (fold change >2 and fdr-*p*-value-corrected ≤ 0.25) (Figure 3) found that five phenolic metabolites are the most important: hesperetin glucuronide, hesperetin sulfate_1, 3.4-dihydroxyphenylacetic acid sulfate_2, hippuric acid glucuronide_1 and hesperetin sulfate_2, with 3,4-dihydroxyphenylacetic acid sulfate_2 being the only one that decreased.

### 2.3. Phenolic Profiles According to the Metabolic Status of the Participants

Volunteer participants were divided according to their metabolic status between C and MetS participants. Considering the total amount of phenolic compounds between groups at baseline, healthy participants showed a higher content of total phenolic compounds (Figure 4).

Interestingly, a PCA analysis did not show statistical differences between the metabolic groups (*p* > 0.05), although according to a posterior fold-change analysis, 15 phenolic compounds were diminished and 46 increased in the MetS group with respect to the control group. This trend continued after the chronic beer consumption under a Mediterranean diet, as the PCA analysis did not show differences between the two groups at the final stage (*p* > 0.05), although the fold change analysis indicated that, at this level, 18 metabolites diminished in the MetS group and 36 increased with respect to the Control group (Figure 5).

Following these initial prospective analyses, we performed a linear model with covariate adjustment to find those phenolic metabolites that differently changed in both groups after the beer intervention, finding that six phenolic metabolites changed differently across the different phenotypes, healthy or MetS. Thus, while hesperetin glucuronide, hesperetin sulfate_1, hesperetin sulfate_2, protocatechuic acid glucuronide and naringerin increased in the C group, they decreased or remained almost unchanged in the MetS group. On the other hand, the 3,4-dihydroxyphenylacetic acid sulfate_2 diminished in the C group, while increasing in the MetS group (Figure 6).

### 2.4. Relationships Between Biochemical Phenolic Profiles According to the Metabolic Status of the Participants

The relationship among the most relevant (poly)phenols seen previously, and their impact on certain biochemical values related to metabolic syndrome, was also evaluated. Significant correlations were found between most of them with the different variables. Furthermore, consideration was given to the (poly)phenols associated with metabolic changes. Those (poly)phenols exhibiting significant opposite or similar correlations for both groups were selected (Figure 7).

The figure presents a bubble plot of the Spearman correlations between different (poly)phenols and metabolic parameters. The initial segment of the figure presents the (poly)phenols highlighted in the previous analysis and various metabolic parameters. As demonstrated in the figure, the (poly)phenols protocatechuic acid glucuronide and 3,4-dihydroxyphenylacetic acid sulfate 2 predominantly exhibit a negative correlation with parameters associated with the anthropometric profile. In addition, naringenin, hesperetin sulfate 1 and 2, and hesperetin glucuronide correlate negatively with other parameters such as insulin, C-peptide, triglycerides and correlate positively with HDL cholesterol and protein parameters. The following section of the figure shows the correlations between another group of (poly)phenols and metabolic parameters, differentiated by group. With the exception of certain (poly)phenols which demonstrate contradictory and substantial correlations according to the metabolic state of the individual, mainly positive correlations are observed between several (poly)phenols such as tyrosol glucuronide, urolithin B glucuronide, vanillic acid and D3,4-dihydroxyphenylacetic acid with metabolic parameters such as Homeostasis Model Assessment of Insulin Resistance (HOMA) index, glucose, BMI, blood pressure, and insulin. In contrast, compounds such as enterodiol glucuronide show negative correlations with anthropometric parameters regardless of the subject’s metabolic status.

## 3. Discussion

Fermented food, and particularly, fermented beverages, have been the focus of extensive research because of their relationships with healthy outcomes. Most of these improvements have been related to their polyphenolic compounds. In this study, we have observed that the chronic consumption of the fermented beverage beer, within a context of a Mediterranean diet, is able to change the urine phenolic compound profile. Moreover, the presence or absence of metabolic syndrome among participants influenced the changes observed in phenolic compound profiles.

Our data clearly indicates that a chronic consumption of beer, equivalent to one bottle per day over a period of six weeks, results in changes to the phenolic compounds present in urine, particularly, suggesting a shift towards a more homogeneous profile. Indeed, (poly)phenol intake has been related to improvements in the metabolic profile [10], so the continued (poly)phenol intake through beer consumption could lead to metabolic changes that affect (poly)phenol levels. Furthermore, our results are similar to those described in the literature, where the circulating forms of the different (poly)phenols are glucuronides and sulfoglucuronides [11]. Thus, this may support a greater homogeneity in the phenolic profile after the intervention, suggesting an overall metabolomic response to regular beer consumption [12].

When we focused on the abundance changes of different phenolic compounds through a fold change analysis, we observed a notable increase, in the whole population, in metabolites primarily derived from citrus flavanones, such as hesperetin glucuronide and hesperetin sulfates, as well as naringenin and its conjugates. These compounds, abundant in beer due to its hop and cereal content [13], could reflect both intestinal absorption and hepatic and microbial metabolism of flavones [14]. The observed increases in dihydroresveratrol sulfate and synaptic acid glucuronide also suggest an effect of gut microbiota on the transformation of stilbenes and phenolic acids [15]. On the other hand, reductions were observed in compounds such as 3,4-dihydroxyphenylacetic acid sulfate, chlorogenic acid, and hydroxytyrosol sulfate and vanillic acid. These decreases could be due to the competition in hepatic conjugation pathways that modulates the catabolism of certain phenols [16].

These results were reinforced by posterior analysis. Indeed, our volcano plot results indicated a selective increase in 3,4-dihydroxyphenylacetic acid sulfate_2, which means an activation of the microbial or host catabolic pathways associated with the degradation of phenolic compounds derived from diverse (poly)phenols and dopamine. These results are relevant since there are preclinical studies conducted in animals that have associated this compound with the prevention of obesity [17]. Concurrently, a consistent decrease in hesperetin conjugates (glucuronide and sulfates) was detected in MetS subjects, which could be due to an increased clearance rate mediated by hepato-renal transporters [18]. Furthermore, the reduction in hippuric acid glucuronide suggests a potential alteration in the gut microbiota responsible for its production [19]. In a clinical context, this profile could be indicative of intestinal dysbiosis and oxidative stress, which is consistent with the metabolic characteristics of the study population [19].

Our study included two specific types of participants, healthy volunteers and MetS patients. In fact, the two groups were clearly differentiated. In particular, MetS patients showed higher body weight and BMI compared to the C group, indicative of a state of overweight or obesity that is characteristic of metabolic syndrome. Furthermore, the subjects with MetS had higher blood pressure, fasting glucose, and lipid profile alterations, which are common in people with metabolic syndrome [20].

Regarding the phenolic compounds, higher concentrations were found in healthy volunteers than in MetS patients. This outcome could be due to several factors, such as better dietary habits or metabolic regulation, which may affect the absorption and metabolism of phenolic compounds [10]. Phenolic compounds are also recognized for their antioxidant and anti-inflammatory properties, and therefore their levels may vary in response to inflammation and oxidative stress as the body tries to protect against these changes by using them as a response [9,10,21]. Thus, taking into account that low-grade inflammation and oxidative stress are present in metabolic syndrome, lower amounts of these compounds are to be expected.

Moreover, differences in the magnitude and direction of phenolic changes were also identified in the two types of participants: healthy subjects showed greater increases, whereas individuals with MetS had attenuated or even decreased responses in the compounds. This finding suggests the potential for alterations in the absorption, conjugation, or excretion pathways of (poly)phenols in individuals with MetS [22]. For example, studies suggest that increases in some of these organic acids may indicate good metabolic function and good gut function, as in the case of hippuric acid [23,24].

Even though PCA revealed no statistically significant differences between the metabolic groups according to their phenolic profiles (*p* > 0.05), a thorough examination revealed significant alterations. Specifically, a decrease in 15 phenolic compounds and an increase in 46 phenolic compounds were observed in the MetS group compared to the C group. The remaining analyses corroborated this finding. This trend was sustained in the post-intervention phase, with no discernible alterations in the overall structure of the PCA (*p* > 0.05). These findings suggest that, although the differences are not reflected in the overall patterns of variance, there are specific modifications at the metabolomic level that could be relevant in the context of metabolic syndrome.

Looking at the six (poly)phenols with the greatest observed differences suggests that a beer intervention supports a higher (poly)phenols intake, but it is the metabolic state of the subject what mainly determines the metabolic pathways. In healthy subjects, the phase II conjugates of hesperetin and naringenin were increased, indicating efficient intestinal absorption and preserved hepatic metabolism. Conversely, within MetS patients, these conjugates remained stable or decreased, suggesting alterations in hepatic detoxification (typical of insulin resistance) and/or increased passage from the aglycone to the colon, where they are degraded by the microbiota before being conjugated. This pattern is consistent with the findings of in vitro studies, which demonstrate that hesperetin-7-*O*-glucoside elevates levels of *Bifidobacterium* and enhances the process of β-deglycosylation. The availability of conjugates is found to be contingent on intestinal integrity [25].

Concurrently, the MetS patients had increased levels of 3,4-dihydroxyphenylacetic sulfate, which is a bacterial phenolic catabolite. This finding indicates that the microbiota present in patients may have the capacity to synthesize simple phenolic acids. However, the loss of conjugates has the potential to compromise vasculoprotective effects [26]. Together, dysbiosis and liver dysfunction in metabolic syndrome appear to shift the balance from active conjugated metabolites to bacterial catabolites.

While most (poly)phenols can improve lipid and glycemic profiles, in many cases, it is the combination of several related (poly)phenols that provide synergistic effects and generate more significant health effects [27]. These synergistic effects may underlie specific metabolic associations observed in clinical studies. Considering the relationship of (poly)phenols with metabolic parameters, the clinical relevance of these changes was reinforced: protocatechuate glucuronide and 3,4-dihydroxyphenylacetic sulfate showed negative correlations with anthropometric markers, while hesperetin and naringenin were positively associated with HDL-cholesterol and negatively with insulin and triglycerides. These findings provide support for the role of beer-derived phenolics as modulators of antioxidant and inflammatory pathways, with potential benefits in improving lipid profiles and insulin sensitivity [8,10,28,29,30].

Moreover, it has been observed that the alterations produced by metabolic syndrome can result in changes in the metabolism of certain (poly)phenols. These alterations can have a positive or negative effect on these parameters, depending on the subject’s condition [31].

Although the current study presents some strengths such as the fact that it has been performed under a registered clinical trial, we also acknowledge some limitations in our study, such as the reduced number of participants or the fact that some of the metabolized (poly)phenols could have originated from dietary sources beyond the beer consumption. However, mitigation activities were introduced, such as different statistical approaches for the first one and a standardization of the diet and follow-up throughout the trial for the second.

## 4. Materials and Methods

### 4.1. Intervention Design and Sample Processing

The present study was conducted at the Hospital Universitario Virgen de la Victoria in Málaga, Spain. A total of 20 adult volunteers, aged between 30 and 60 years, were recruited to participate in a randomized, crossover study with an open-label design (ClinicalTrials.gov identifier NCT05300165). The main outcome was previously published elsewhere [9]. Briefly, in order to be eligible for participation in the trial, individuals were required to have a BMI of <40 kg/m^2^ and to abstain from the consumption of more than 20 g and 30 g of alcohol per day for women and men, respectively. The subjects were divided into two groups: a control group (C) (n = 10) comprising subjects with a BMI of <30 kg/m^2^, and a group of subjects with metabolic syndrome (MetS) (n = 10), defined according to the National Cholesterol Education Program Adult Treatment Panel III criteria [32].

The participants were randomly assigned to a crossover design, after a two-week period of washout, in which the participants were trained to follow a Mediterranean diet, which was maintaining during the whole study. Particularly, the participants were instructed to consume olive oil as the primary source of fat, as well as a diet rich in fruits, vegetables, legumes, nuts, whole grains, as well as a moderate consumption of fish, seafood, poultry, eggs, and a low consumption of red and processed meats. These patterns were distributed in 3 main meals (breakfast, lunch, and dinner), and an additional 2 snacks periods.

Participants were instructed to consume one bottle (330 mL) of beer per day for a total period of six weeks. The quantity supplied guaranteed that the maximum daily alcohol intake did not exceed 12 g. During the intervention, participants were permitted to consume only the beers listed in the study protocol, and no other alcoholic beverage. Information relative to the kind of beer consumed can be found in our previous manuscript [9], but briefly: all participants consumed three types of beer for a period of two weeks each one: an alcohol-free beer (low (poly)phenols content—12.2 mg/100 mL); a lager beer (intermediate (poly)phenols content—27.83 mg/100 mL; 4.2% alcohol by volume); and a dark beer (high (poly)phenols content—41.6 mg/100 mL; 4.5% alcohol by volume). (Poly)phenols content was estimated using the public database Phenol-Explorer [33]. Beer consumption was allocated into a randomized model mitigating the possible interference due to the different sequences performed. Anthropometric, biochemical, and dietary assessments were conducted at baseline and final time-points (after 6 weeks of a daily beer bottle consumption). Subjects were evaluated at the beginning of the day; which was the time-point when all measurements were taken. Furthermore, a urine sample was collected and immediately frozen at −80 °C until subsequent analysis. In addition, a dietary survey was completed by each subject to facilitate the monitoring of their dietary habits. Blood samples were collected after 10 h of fasting. Weight, height, and waist and hip circumferences were measured according to standardized procedures and BMI was calculated as weight (kg) divided by height^2^ (m^2^). Biochemical parameters including total cholesterol (mg/dL), HDL cholesterol (mg/dL), triglycerides (mmol/L), and serum glucose were measured using a standard enzymatic method (Randox Laboratories Ltd., Crumlin, UK). Low-density lipoprotein (LDL) cholesterol (mg/dL) was calculated by using the Friedewald formula.

The informed consent of all subjects participating in the study was obtained. The study was conducted in accordance with the ethical guidelines established in the Declaration of Helsinki and received approval from the Ethics Committee of the Hospital Universitario Virgen de la Victoria (reference ID BEEROTA18).

### 4.2. Phenolic Compound Analysis

The analysis of profiles of phenolic compounds was performed using urine samples obtained from the subjects participating in the study. Urine was selected as the matrix for the analyses due to its common employment in the study of (poly)phenols metabolism, as it reflects the absorbed and metabolized forms of these compounds. The excretion of these compounds primarily occurs via the urine, and the cumulative excretion of these substances provides a more accurate evaluation of absorption than plasma levels, given that urinary concentrations are typically higher [34].

Sample preparation and analysis were conducted following a previously described protocol [35], with minor modifications to expand the method for the comprehensive analysis of urinary phenolic metabolites. The samples were subjected to a dilution process involving the addition of 10 µL of urine to 80 µL of water (with the presence of 0.05% formic acid), along with 10 µL of internal standard. The final concentration was of 100 µg/L. The internal standards utilized included hippuric-d5, abscisic acid-d6, hydroxytyrosol-d4, vanillic-d3 acid, and enterodiol-d6, with a final concentration of 100 µg/L. A standard curve was constructed in water at concentrations ranging from 1 to 7500 ppb (µg/L) for the performance of a quantitative analysis. The following compounds were used: apigenin; caffeic acid; caffeic glucuronide 3-hydroxyphenylacetic acid; 3-hydroxybenzoic acid; 3,4-dihydroxyphenylacetic acid; 4-hydroxybenzoic acid; ferulic acid; fisetin; genistein; hesperetin; hesperidin; catechin; epicatechin; epicatechin gallate; gallic acid; chlorogenic acid; daidzein; dihydroresveratrol; elenolic acid; enterodiol, and enterolactone. The samples were subsequently analyzed by high-performance liquid chromatography coupled to high-performance mass spectrometry (HPLC-ESI-QTOF-HRMS), at the Separation Techniques Unit of the Scientific and Technological Centers (CCiTUB). Chromatographic separation was performed on an Agilent 1290 Infinity II (Agilent Technologies, Santa Clara, CA, USA) using a Kinetex F5 column coupled with a SecurityGuard™ UHPLC F5 precolumn (Phenomenex, Torrance, CA, USA) were utilized, with water and acetonitrile as the mobile phases, both containing 0.05% acetic acid. The mass spectrometry was conducted using an Agilent 6560 QTOF-HRMS instrument, with the polarity set to negative. The identification and quantification of phenolic metabolites were carried out using Agilent MassHunter Workstation Software (Qualitative Analysis Version 10.0 and Quantitative Analysis Version 10.0) (Agilent Technologies, Santa Clara, CA, USA). Compounds were identified by comparing retention times and MS/MS spectra with those of commercial standards, when available. If not, identification was based on their fragmentation patterns obtained through product ion scans. Quantification was performed using calibration curves prepared in synthetic urine. Standard curves were constructed at concentrations ranging from 1 to 7500 ppb (µg/L). Each compound was quantified in relation to its pre-established standard. In the absence of a standard, an analogous alternative was selected based on chemical similarity to its most relative compound within the standards, i.e., hydroxytyrosol sulfate was quantified with the hydroxytyrosol curve.

### 4.3. Statistical Analysis

The statistical analysis of the results was conducted using SPSS 22.0 (SPSS Inc., Chicago, IL, USA), R 4.4.2 (www.r-project.org), and Python 3.12.1 (www.python.org). The more complex analyses of the phenolic compounds were conducted using MetaboAnalyst 6.0, which offers a comprehensive range of methods for the analysis and visualization of metabolomics data [36]. In particular, data was primarily analyzed through paired samples after normalization of the concentrations, particularly the PCA, Fold change, and Volcano plot utilities from the statistical analysis (one factor), and after that, the linear model analysis from the statistical analysis (metadata table) between groups and times.

Moreover, in selected cases, outliers were excluded prior to conducting the statistical analyses. In the majority of cases, non-parametric methods were utilized. The Mann–Whitney U test was employed to evaluate the independent variables and identify any significant differences between the participant groups (C and MetS). Furthermore, PCA, t-SNE, Isomap and PLS regression were employed as analytical techniques. In addition, the procrustes analysis was performed to evaluate the degree of dissimilarity between sample configurations. Statistically significant differences were identified (*p* < 0.05).

## 5. Conclusions

The present findings show that beer-derived (poly)phenols are metabolized in a differential way depending on the metabolic health status of the subjects. This outcome suggests that the variability in metabolic profile may be attributed to differences in enzyme activity, gut microbial composition, or genetic factors that modulate the absorption, metabolism or biotransformation of (poly)phenols.

Furthermore, these bioactive compounds may exert regulatory effects on specific metabolic pathways associated with oxidative stress, systemic inflammation, and lipid metabolism. These results provide support for the hypothesis that beer-derived (poly)phenols may contribute to certain metabolic health benefits by acting as the modulators of specific biochemical processes, thereby conferring them a certain degree of therapeutic potential. Nevertheless, it is imperative to exercise caution and avoid making recommendations regarding consumption levels, due to the damaged effects of certain compounds, such as an excess of alcohol.

## Figures and Tables

**Figure 1 molecules-30-02932-f001:**
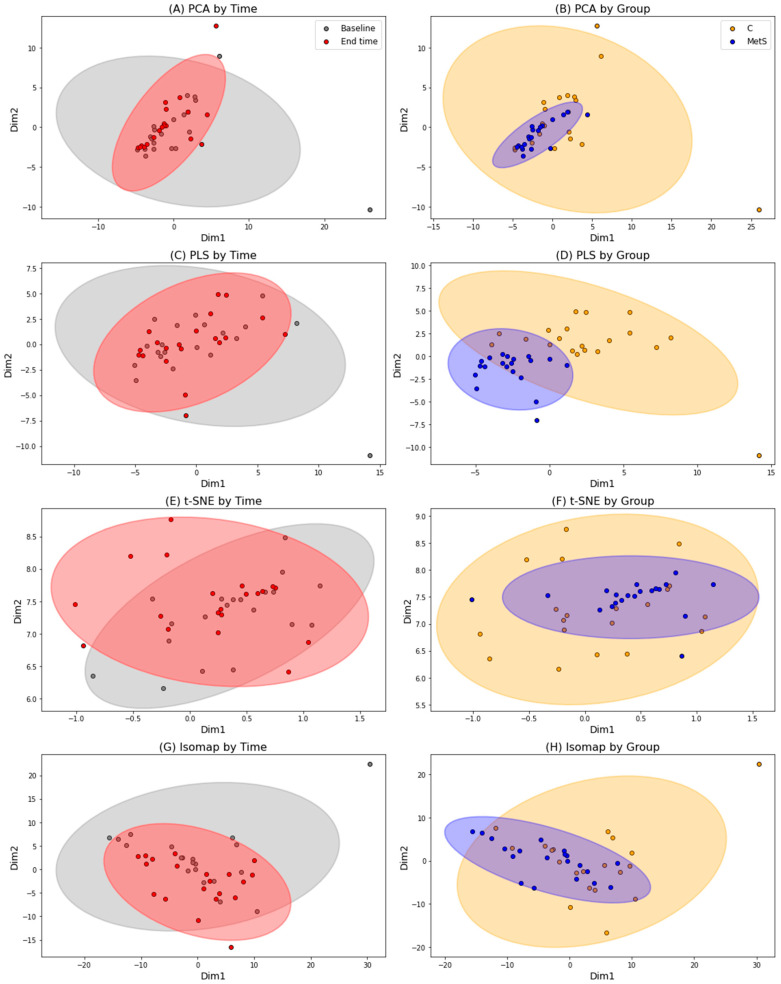
Comparative visualization of PCA, PLS, t-SNE, and isomap by group and time. Color legend: grey = baseline; red = end time; orange: MetS group; blue: Control group.

**Figure 2 molecules-30-02932-f002:**
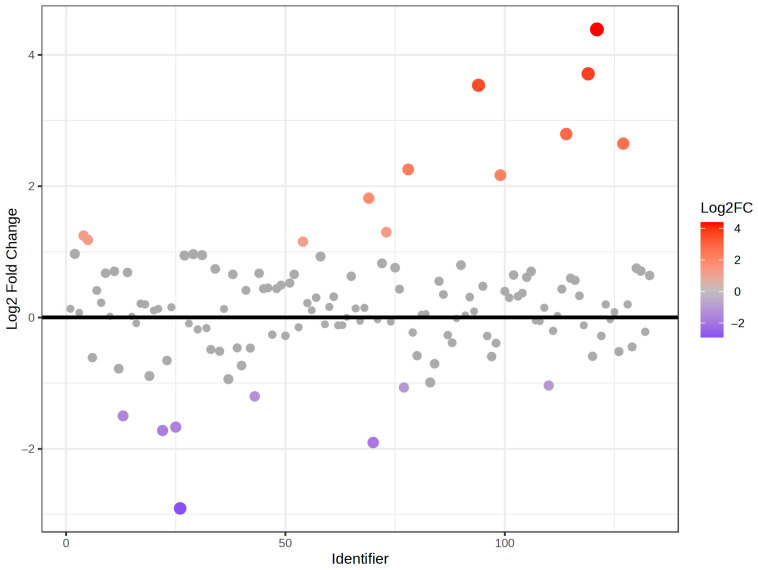
Fold change analysis of phenolic metabolites after the beer intervention in the whole population of study.

**Figure 3 molecules-30-02932-f003:**
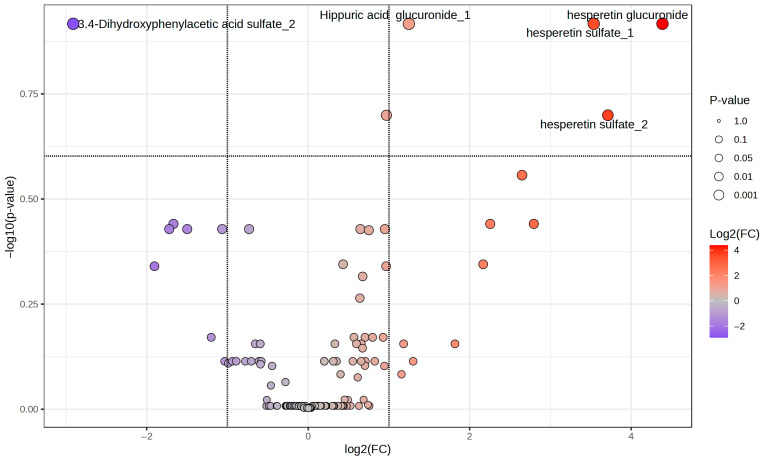
Volcano plot of the phenolic compounds that changed with the beer intervention in the whole population.

**Figure 4 molecules-30-02932-f004:**
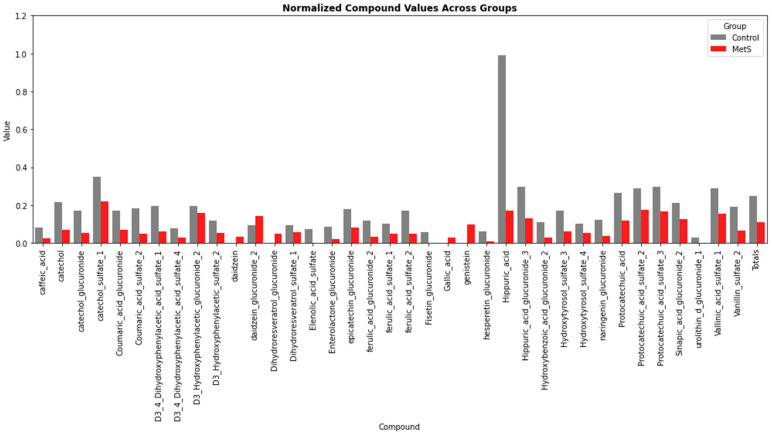
Difference in the concentration of phenolic compounds according to subject groups at baseline.

**Figure 5 molecules-30-02932-f005:**
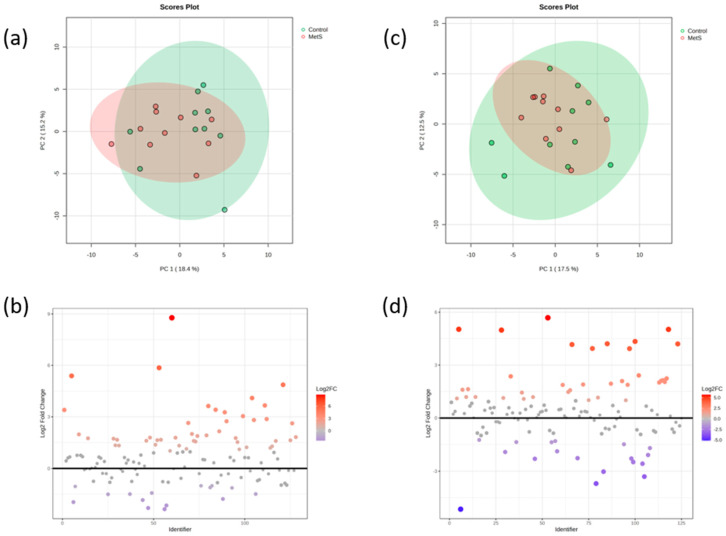
Basal and final patterns of phenolic compounds between the control group and metabolic syndrome patients (MetS). (**a**) Representation of the PCA analysis of phenolic compounds at baseline between MetS and Control groups; (**b**) Fold change analysis of phenolic compounds at baseline between MetS and Control groups; (**c**) Representation of the PCA analysis of phenolic compounds at a final point between the MetS and Control groups; (**d**) Fold change analysis of phenolic compounds at a final point between the MetS and Control groups.

**Figure 6 molecules-30-02932-f006:**
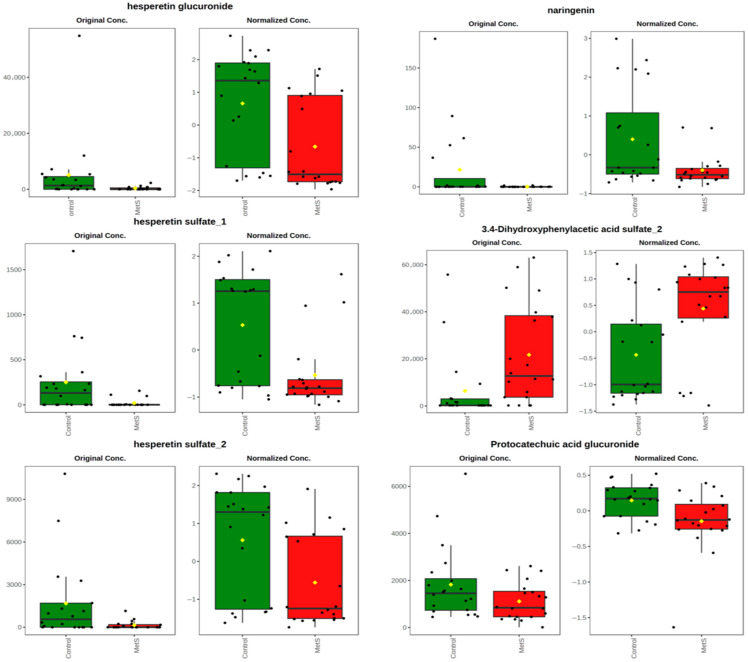
Phenolic compounds that changed differently among study groups according to the linear models results. The figure shows changes in concentration between times and groups. Control group: green; metabolic syndrome (MetS) patients: red. The yellow diamond represents the group mean value.

**Figure 7 molecules-30-02932-f007:**
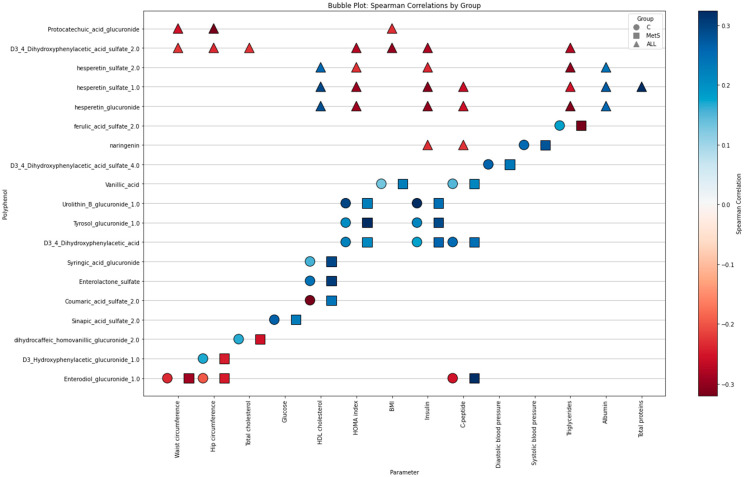
Relationship between (poly)phenols and metabolic parameters: Spearman’s correlations by group and totals of relevant (poly)phenols. Subject groups: circle = Control, square = MetS, triangle = All subjects. Blue for positive correlations and red for negative correlations.

**Table 1 molecules-30-02932-t001:** Mean and standard deviation of selected anthropometric, biochemical and nutritional parameters.

Parameters (Control n = 10/MetS n = 10)	Control—Baseline	Control—End Time	MetS—Baseline	MetS—End Time
Weight (kg) ^a,b^	65.6 ± 7.8	65.4 ± 7.4	93.7 ± 14.4	93.1 ± 15.1
BMI (kg/m^2^) ^a,b^	22.9 ± 2.5	22.8 ± 2.4	34.6 ± 4.8	34.3 ± 4.9
Waist (cm) ^a,b^	80.3 ± 8.9	79.6 ± 7.5	109.1 ± 10.2	104.9 ± 8.5
Hip (cm) ^a,b^	97.0 ± 6.4	97.0 ± 6.6	114.9 ± 12.8	112.6 ± 12.4
Systolic blood pressure (mmHg) ^a,b^	114.0 ± 14.5	112.1 ± 12.4	137.5 ± 13.2	129.4 ± 12.1
Diastolic blood pressure (mmHg) ^a,b^	69.4 ± 8.3	71.1 ± 10.1	88.8 ± 10.3	86.7 ± 6.7
Glucose (mg/dL) ^a,b^	82.6 ± 6.8	88.5 ± 5.8	114.7 ± 36.2	119.0 ± 24.6
Creatinine (mg/dL)	0.8 ± 0.1	0.8 ± 0.1	0.7 ± 0.2	0.8 ± 0.2
Uric Acid (mg/dL) ^b^	4.2 ± 0.8	4.5 ± 1.1	5.3 ± 1.4	5.9 ± 1.3
Na (mEq/L)	140.0 ± 1.3	140.5 ± 2.2	140.3 ± 2.2	139.0 ± 3.7
Fe (µg/dL)	77.5 ± 20.4	75.6 ± 24.2	71.1 ± 34.4	62.1 ± 27.6
Cholesterol (mg/dL)	192.5 ± 31.0	190.1 ± 30.3	208.6 ± 43.1	214.4 ± 45.4
HDL (mg/dL) ^a,b^	64.0 ± 12.6	65.9 ± 12.4	48.7 ± 11.7	47.9 ± 12.1
LDL (mg/dL)	113.6 ± 28.6	109.6 ± 27.8	129.6 ± 31.2	127.9 ± 39.3
Triglycerides (mg/dL) ^a,b^	74.6 ± 38.6	73.1 ± 33.5	151.7 ± 89.2	206.4 ± 168.2
Energy (kcal)	1610.1 ± 291.6	1704.3 ± 203.3	1617.2 ± 549.3	1697.6 ± 297.8
Total Protein (g)	65.7 ± 15.2	68.0 ± 13.0	71.0 ± 34.0	74.1 ± 14.0
Total Fat (g)	81.7 ± 24.4	79.6 ± 16.8	82.4 ± 38.7	89.9 ± 22.0
Total Carbohydrates (g)	155.3 ± 26.8	178.4 ± 27.6	147.8 ± 44.8	147.3 ± 29.3
Dietary Fiber (g)	16.6 ± 6.4	16.1 ± 12.2	14.6 ± 5.0	14.3 ± 5.7

Values are presented as mean ± SD (standard deviation). *p* values were calculated for differences between groups using a Mann–Whitney U test. Statistically significant differences (*p* < 0.05) at both baseline (^a^) and end time (^b^).

## Data Availability

The data presented in this study are available under request from the corresponding author due to ethical reasons.

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
