# Peer review of "Beer-Derived (Poly)phenol Metabolism in Individuals With and Without Metabolic Syndrome: A Comparative Dietary Intervention"

_molecules, 2025, doi:10.3390/molecules30142932_

Round 1
Reviewer 1 Report
Comments and Suggestions for Authors
The manuscript presents research on beer consumption over a six-week period, in the context of a Mediterranean diet, investigating how beer-derived polyphenols are metabolized differently in individuals with and without metabolic syndrome. The results obtained by the authors highlighted significant variations in the urinary profiles of phenolic compounds. Healthy participants showed greater increases in polyphenol levels, indicating more efficient absorption and metabolism, while individuals with metabolic syndrome exhibited attenuated metabolic responses, possibly due to dysbiosis and liver dysfunction. The authors suggest that these changes indicate a complex interaction between beer polyphenols, gut microbiota, and the individual's metabolic state, with potential impact on metabolic and inflammatory markers. However, caution is needed in interpreting these effects due to the alcohol content of beer.
My observations are as follows:
- Could some of the metabolized polyphenols have originated from dietary sources?
- Perhaps it would have been better if the gender ratio were reversed.
- In Table 1, "Mets" should be corrected to "MetS". Additionally, the nutritional parameters listed should be explained more clearly.
- Line 126 – PLS should be defined, just as PCA was in line 118, even if these abbreviations are familiar to many readers.
Author Response
First of all, the authors would like to thank the anonymous reviewer for the helpful and constructive comments that greatly contributed to improving the final version of the manuscript. The authors have tried to follow and answer all the recommendations and commentaries of the reviewer. The answers of the authors have been indicated by “AU”. The changes within the manuscript have been highlighted in red font in the revised manuscript. Authors hope the revision will meet the demands of the reviewer and Editor.
The manuscript presents research on beer consumption over a six-week period, in the context of a Mediterranean diet, investigating how beer-derived polyphenols are metabolized differently in individuals with and without metabolic syndrome. The results obtained by the authors highlighted significant variations in the urinary profiles of phenolic compounds. Healthy participants showed greater increases in polyphenol levels, indicating more efficient absorption and metabolism, while individuals with metabolic syndrome exhibited attenuated metabolic responses, possibly due to dysbiosis and liver dysfunction. The authors suggest that these changes indicate a complex interaction between beer polyphenols, gut microbiota, and the individual's metabolic state, with potential impact on metabolic and inflammatory markers. However, caution is needed in interpreting these effects due to the alcohol content of beer.
My observations are as follows:
Could some of the metabolized polyphenols have originated from dietary sources?
AU: We appreciate the comment of reviewer. As the reviewer correctly asserts, metabolites can indeed originate from the diet. However, a thorough analysis, as outlined in line 104 of the manuscript “It is noteworthy that no statistically significant differences in nutritional parameters were identified during the study.”, suggests that fluctuations in metabolite levels are predominantly influenced by beer consumption. Moreover, in line with other reviewers’ requests, we have added further information relative to the dietary patterns followed by the volunteers during the study. However, the authors acknowledge the worries about this issue. We have included further explanation in material and methods regarding the continuation of the diet during the whole study, as well as we have introduced this fact into a new paragraph about strengths and limitations of the study.
Perhaps it would have been better if the gender ratio were reversed.
AU: We thank the reviewer for the comments, the gender ratio has been changed in the new version of the manuscript.
In Table 1, "Mets" should be corrected to "MetS". Additionally, the nutritional parameters listed should be explained more clearly.
AU: In response to the reviewer's request, the table has been updated and an explanation of the nutrients has been incorporated into Line 106. “Furthermore, the principal values obtained for dietary intake with regard to energy and macronutrient consumption are presented in Table 1”.
Line 126 – PLS should be defined, just as PCA was in line 118, even if these abbreviations are familiar to many readers.
AU: We appreciate the reviewer's comments and would like confirm that PLS has been defined (line 127), in the new version of the manuscript.
Reviewer 2 Report
Comments and Suggestions for Authors
The work is relevant and brings relevant information to the area. However, some information must be included in the manuscript.
The introduction is relevant and provides essential information.
1- Lina 90 says, " This study was a randomized open-label crossover trial." From my understanding, there were 2 groups: a control and a MetS. That is? And they all got beer. Correct? After two weeks, they received a Mediterranean diet. So everyone did the beer and Mediterranean diet for 6 weeks? Explain better, since the crusade would be first drinking beer and then changing the treatment, which would be 2 treatments, with a washout period between the two treatments.It is confused.
2-If a specific diet has been prepared, add this data.
3- Add the brand of beer and, if possible, the quantification of the phenolic compounds present in the sample.
4- In Table 01, we have various information about the research participants. Add in the methods how each result was obtained (methodology applied for each result).
5-Still in table 01, add the n used.
6-Add the chromatograms
7-Detail the technique for identifying and quantifying phenolic compounds
8-Line 313: space between in15
9- in vitro should be in italic
Author Response
First of all, the authors would like to thank the anonymous reviewer for the helpful and constructive comments that greatly contributed to improving the final version of the manuscript. The authors have tried to follow and answer all the recommendations and commentaries of the reviewer. The answers of the authors have been indicated by “AU”. The changes within the manuscript have been highlighted in red font in the revised manuscript. Authors hope the revision will meet the demands of the reviewer and Editor.
The work is relevant and brings relevant information to the area. However, some information must be included in the manuscript.
The introduction is relevant and provides essential information.
AU: We would like to express our gratitude to the reviewer with regard to their positive comments.
1- Lina 90 says, " This study was a randomized open-label crossover trial." From my understanding, there were 2 groups: a control and a MetS. That is? And they all got beer. Correct? After two weeks, they received a Mediterranean diet. So everyone did the beer and Mediterranean diet for 6 weeks? Explain better, since the crusade would be first drinking beer and then changing the treatment, which would be 2 treatments, with a washout period between the two treatments. It is confused.
AU: The authors apologize for this insufficient information at this point of the manuscript, maybe related to the fact that the design is developed into the material and methods section beyond this point. Thus, in order to improve the comprehension of the study at this point, we have included further information. But as the reviewer correctly states, two different types of volunteers, people with and without MetS were investigated about the impact of beer consumption on their urine (poly)phenol profile. In order to standardize their diet, they performed an initial two weeks Mediterranean diet, which was continued during the 6 weeks of the intervention study. The polyphenol profile was assessed at baseline (after two weeks of a Mediterranean diet and without beer consumption) and at the end of the study (after 6 weeks of a Mediterranean diet plus beer consumption).
2-If a specific diet has been prepared, add this data.
AU: The authors thanks the reviewer’s comment, nevertheless, it should be noted that the subjects did not adhere to a completely controlled diet; they were trained to follow a Mediterranean diet which was monitored throughout the whole study every two weeks. This permitted to normalize the diet of the volunteers. More information relative to this Mediterranean pattern has been introduced in the Material and Methods section.
3- Add the brand of beer and, if possible, the quantification of the phenolic compounds present in the sample.
AU: The authors appreciate this comment of the reviewer, and apologize for insufficient information about this important item. We have introduced more information regarding the type of beers used as well as their polyphenol and alcohol contents. “Information relative to the kind of beer consumed can be found in our previous manuscript [20], but briefly: all participants consumed three types of beer for a period of two weeks each one: an alcohol-free beer (low polyphenol content—12.2 mg/100 mL); a lager beer (intermediate polyphenol content—27.83 mg/100 mL; 4.2% alcohol by volume); and a dark beer (high polyphenol content—41.6 mg/100 mL; 4.5% alcohol by volume); polyphenol content was estimated using the public database Phenol-Explorer [21]. Beers consumption was allocated into a randomized model mitigating the possible interference due to different sequences performed”. However, the brand of beer cannot be introduced, as it has been stated in the acknowledgments section, they were donated by FICYE.
4- In Table 01, we have various information about the research participants. Add in the methods how each result was obtained (methodology applied for each result).
AU: We would like to thank the reviewer for their comments, which have enabled us to make improvements to the manuscript. As suggested by the reviewer we have added in the Material and methods section more information relative to the methods followed to measure the variables included in table 1.
5-Still in table 01, add the n used.
AU: In response to the reviewer's request, the table has been updated
6-Add the chromatograms
AU: Thanks for your suggestion. However, the authors consider that the chromatograms do not add relevant information to the main aim of the manuscript that is to investigate the differences in the (poly)phenols content with the beer intervention as well as the differences into different metabolic status of the patients. 40 urine samples were analyzed and one random chromatogram would not add relevant information. The section about the phenolic compound analysis includes the pertinent information regarding the analysis, which has been expanded according to the following request of the reviewer.
7-Detail the technique for identifying and quantifying phenolic compounds
AU: Thanks for the suggestion. The section about the phenolic compound analysis have been increased in order to introduce further information relative to the identification and quantification of the phenolic compounds.
8-Line 313: space between in15
AU: We would like to express our gratitude to the reviewer for highlighting this issue, which has now been rectified.
9- in vitro should be in italic
AU: We would like to express our gratitude to the reviewer for highlighting this issue, which has now been rectified.
Reviewer 3 Report
Comments and Suggestions for Authors
This article explores how beer consumption impacts polyphenol metabolism in individuals with and without metabolic syndrome (MetS). It carries significance in offering potential dietary strategies for MetS prevention and management through its innovative approach. Despite limitations such as a small sample size and short intervention period, the findings are significant for advancing our understanding of polyphenols in metabolic health. With some minor revisions to address the following limitations, it would be accepted by the journal "Molecules."
- There are spelling and grammatical errors. For instance, at line 37, “ (pol)yphenol-derived” should be “(poly)phenol-derived”; at line 81, it is better to change “MetS parameter” to “MetS parameters”. The terms “(poly)phenols” and “polyphenols” are used inconsistently and it is recommended to unify them.
- In Table 1, to maintain consistency in the number of significant digits (the number of digits after the decimal point) for the data.
- The results presented by Isomap and t-SNE in Figure 1 are not described and analyzed in the text. Please add the relevant analysis.
- At line 155-158, the t-test analysis only describes the substances with differences between groups, but lacks statistical significance analysis. It is suggested to add p-values or confidence intervals.
- Line 202-204, the description of the content changes of protocatechuic acid glucuronide in the figure is inaccurate. In group C, the content of protocatechuic acid glucuronide should be increased.
- In section “4.2 Phenolic compound analysis”, it is suggested to supplement references to support the method of studying polyphenol metabolism using only urine samples as there may be limitations to this approach.
There are some spelling and grammatical errors.
Author Response
First of all, the authors would like to thank the anonymous reviewer for the helpful and constructive comments that greatly contributed to improving the final version of the manuscript. The authors have tried to follow and answer all the recommendations and commentaries of the reviewer. The answers of the authors have been indicated by “AU”. The changes within the manuscript have been highlighted in red font in the revised manuscript. Authors hope the revision will meet the demands of the reviewer and Editor.
This article explores how beer consumption impacts polyphenol metabolism in individuals with and without metabolic syndrome (MetS). It carries significance in offering potential dietary strategies for MetS prevention and management through its innovative approach. Despite limitations such as a small sample size and short intervention period, the findings are significant for advancing our understanding of polyphenols in metabolic health. With some minor revisions to address the following limitations, it would be accepted by the journal "Molecules."
AU: We would like to express our gratitude to the reviewer with regard to their positive comments.
There are spelling and grammatical errors. For instance, at line 37, “ (pol)yphenol-derived” should be “(poly)phenol-derived”; at line 81, it is better to change “MetS parameter” to “MetS parameters”. The terms “(poly)phenols” and “polyphenols” are used inconsistently and it is recommended to unify them.
AU: We would like to express our gratitude to the reviewer for highlighting these issues. These details have been corrected and improved in the new version of the manuscript.
In Table 1, to maintain consistency in the number of significant digits (the number of digits after the decimal point) for the data.
AU: The authors apologize for this format error. The table 1 has been appropriately corrected in order to standardize the number of decimals shown.
The results presented by Isomap and t-SNE in Figure 1 are not described and analyzed in the text. Please add the relevant analysis.
AU: We appreciate the reviewer's comment. In the new version of the manuscript, we reference these results (line 128).
At line 155-158, the t-test analysis only describes the substances with differences between groups, but lacks statistical significance analysis. It is suggested to add p-values or confidence intervals.
AU: Thanks for the suggestions. All p-values, and their FDR corrected values have been included for further clearance.
Line 202-204, the description of the content changes of protocatechuic acid glucuronide in the figure is inaccurate. In group C, the content of protocatechuic acid glucuronide should be increased.
AU: Thanks for the observation of this mistake. As the reviewer has indicated, protocatechuic acid glucuronide increased in the C group. Changes have been accordingly included within the text.
In section “4.2 Phenolic compound analysis”, it is suggested to supplement references to support the method of studying polyphenol metabolism using only urine samples as there may be limitations to this approach.
AU: We appreciate the reviewer's comment. In the new version of the manuscript, this section has been completed, with the justification for the utilization of urine samples now clearly highlighted. Furthermore, two associated references have been included.
- Marhuenda-Muñoz, M.; Laveriano-Santos, E.P.; Tresserra-Rimbau, A.; Lamuela-Raventós, R.M.; Martínez-Huélamo, M.; Vallverdú-Queralt, A. Microbial Phenolic Metabolites: Which Molecules Actually Have an Effect on Human Health? Nutrients 2019, 11, 2725, doi:10.3390/nu11112725.
- Campins-Machado, F.M.; Casas, R.; Lamuela-Raventós, R.M.; Galkina, P.; Martínez-González, M.Á.; Fitó, M.; Ros, E.; Estruch, R.; Domínguez-López, I.; Pérez, M. Microbiota-Derived Resveratrol Metabolites: New Biomarkers of Red Wine Consumption Are Inversely Associated with Inflammation in a Longitudinal Study of a Mediterranean Population. The Journal of nutrition, health and aging 2025, 29, 100542, doi:10.1016/j.jnha.2025.100542.